# Factor Decorrelation Enhanced Data Removal from Deep Predictive Models

**Wenhao Yang**
Wuhan University of Technology, China
342471@whut.edu.cn

**Lin Li***
Wuhan University of Technology, China
cathylilin@whut.edu.cn

**Xiaohui Tao**
University of Southern Queensland
Queensland, Australia
Xiaohui.Tao@unisq.edu.au

**Kaize Shi**
University of Southern Queensland
Queensland, Australia
Kaize.Shi@unisq.edu.au

## Abstract

The imperative of user privacy protection and regulatory compliance necessitates sensitive data removal in model training, yet this process often induces distributional shifts that undermine model performance-particularly in out-of-distribution (OOD) scenarios. To address this issue we propose a novel data removal approach that enhances deep predictive models through factor decorrelation and loss perturbation. Our approach introduces: (1) a discriminative-preserving factor decorrelation module employing dynamic adaptive weight adjustment and iterative representation updating to reduce feature redundancy and minimize inter-feature correlations. (2) a smoothed data removal mechanism with loss perturbation that creates information-theoretic safeguards against data leakage during removal operations. Extensive experiments on five benchmark datasets show that our approach outperforms other baselines and consistently achieves high predictive accuracy and robustness even under significant distribution shifts. The results highlight its superior efficiency and adaptability in both in-distribution and out-of-distribution scenarios.

## 1 Introduction

Removing specific data in the machine learning model training process is crucial to protect user privacy and regulatory compliance [1, 2, 3]. For example, users of e-commerce platforms may invoke data deletion rights for product reviews that have been incorporated into the training corpus of recommendation models. Satisfying such requests entails removing the associated entries from front-end systems while also ensuring that the date's influence is purged from the model's internal representations and parameter space. In addition, financial clients can request the removal of transaction histories or loan application records that have contributed to the training of credit scoring models. These scenarios highlight that data removal requests are distributed in widely scenarios with domain-specific regulatory and operational constraints. Furthermore, such removals can induce shifts in the underlying data distribution, while retraining the overall model for each of the specific cases is impractical since the computational costs [4, 5]. Therefore, generalizable approaches to data removal are essential for adapting to varied and evolving deletion demands.

The primary challenge confronting data removal methodologies lies in the inadequate exploration and adaptation to out-of-distribution (OOD) data scenarios [6, 7]. Existing data removal mechanisms

---

*Corresponding author. Email: cathylilin@whut.edu.cn

39th Conference on Neural Information Processing Systems (NeurIPS 2025).

predominantly rely on gradient-based updates and parameter fine-tuning, assuming that the data distribution is similar before and after removal, which inherently limits their robustness. As data distributions may evolve dynamically across temporal and contextual dimensions with continuous data removal requests, the intrinsic correlation between feature representations and corresponding labels transforms. These distributional shifts weaken the effectiveness of existing forgetting mechanisms, reduce removal accuracy, and decrease generalization ability to unseen data. Consequently, existing removal mechanisms struggle to maintain model performance under dynamic scenarios.

Feature dimensionality reduction serves as an effective decorrelation strategy in OOD scenarios and has been widely adopted in predictive modelling techniques, including principal component analysis (PCA) [8, 9], clustering-based approaches [10], and kernel-based mappings [11]. For instance, Stablenet [12] employs Random Fourier features to achieve spatial transformation for classification under OOD conditions. A core challenge lies in integrating dimensionality reduction with existing data removal strategies to design parameter update algorithms to maintain the balance between model accuracy and computational efficiency. While dimensionality reduction reshapes the representation space, it may also discard informative and discriminative features as dimensionality decreases. Loss functions without appropriate adaptation may lead to gradient directions that diverge from the true optimization objective, introducing training instability and degrading generalization performance.

To address the aforementioned challenge, we propose DecoRemoval, a data removal method that avoids retraining under OOD scenarios. In such settings, we introduce a discriminative maintenance factor decorrelation module and use a spatial mapping strategy to efficiently reduce feature dimensionality with linear computational complexity. This transformation is based on the Fourier transform of a kernel function, thereby reducing feature redundancy and promoting factor decorrelation. DecoRemoval maintains feature weights through iterative gradient updates, which accelerates convergence and enhances robustness without assuming a fixed data distribution. To further improve the safety and reliability of the removal process, we introduce a random linear perturbation module for smoothed data removal. This perturbation serves as a regularizer in the parameter space, smoothing the solution landscape of the objective function. As a result, it enables accurate approximation of retraining effects via localized parameter adjustments. Compared with several baselines, the proposed DecoRemoval achieves the best performance in balancing accuracy and efficiency in data removal scenarios. The contributions of this paper can be summarized as follows:

- We propose DecoRemoval, a discriminative-preserving factor decorrelation method that integrates feature dimensionality reduction with data removal, which dynamically adjusts feature weights to balance removal precision and computational efficiency.
- We design a smoothed data removal mechanism incorporating a Loss Perturbation module, which introduces linear interference to protect sensitive information while preserving model stability during the removal process.
- We conduct extensive experiments on standard benchmarks, showing that DecoRemoval achieves competitive predictive performance, robust generalization, and high efficiency under significant distributional shifts.

## 2 Related Work

### 2.1 Machine Unlearning

Machine unlearning has emerged as a pivotal area of research in response to growing privacy concerns and regulatory mandates [13, 14, 15, 16]. This field focuses on developing methodologies that enable machine learning models to effectively remove the influence of specific data points without necessitating complete retraining [17, 18].

As concerns about data privacy and regulatory compliance continue to grow, the ability of machine learning models to "forget" specific data points has emerged as a key area of research. Machine forgetting aims to remove the influence of individual data without retraining the entire model [19]. Early approaches include using gradient vectors or summary layers to isolate and mitigate data influence. Existing methods for forgetting in deep neural networks can be broadly categorized into two groups: retraining-based and retraining-free approaches. Retraining-based methods involve re-optimizing the model after data removal, while retraining-free methods avoid this by estimating the sensitivity of model parameters [20, 21]. These methods often rely on approximations using

the Fisher information matrix or the Hessian matrix, as seen in early techniques such as Certified Removal and Optimal Brain Damage [22, 23].

A major challenge in this space has been adapting these forgetting methods to the complex, nonconvex landscape of deep neural networks [24, 25]. To address this, Zhang et al. (2024) extend certified unlearning to deep models by bounding the error introduced by a Newton update, enabling scalable and theoretically grounded forgetting in nonconvex settings [26]. Building on this direction, Foster et al. (2024) introduce Selective Synaptic Dampening, a method that identifies parameters most relevant to the forget set using Fisher information and proportionally reduces their impact [27]. This allows the model to unlearn specific data while maintaining performance on the remaining dataset.

However, in the dynamic environment, data distribution will evolve over time. The existing machine unlearning methods face serious limitations when dealing with scenarios where data distribution is changing. Our method introduces the feature decoupling module into deep neural networks, providing a balance between efficiency and adaptability in both in distribution and out of distribution settings.

## 2.2 Certified Data Removal

Certified data removal methods allow models to "forget" specific data points while maintaining statistical equivalence to models trained without the removed data [28, 29, 30, 31]. The requirement for data removal speed in practical application scenarios of machine learning cannot be ignored. Certified removal stands out for providing a favorable balance between removal speed and accuracy [13]. It can ensure removal accuracy to a certain extent while maintaining extremely high practical efficiency, making it a leading SOTA method in current research literature.

Certified data removal typically adjusts model parameters by removing residual influences of removed samples, often through gradient-based updates and calibrated noise injection [32, 31, 33]. Marchant et al. [14] pioneered a verification framework for unlearning by analyzing the Hessian matrix of training data and gradients associated with removable samples. Their method triggers retraining if theoretical error bounds exceed predefined thresholds. Subsequent work by Neel et al. [15] introduced regularized and distributed gradient descent variants, providing formal guarantees on model indistinguishability and accuracy for weakly convex loss functions. Guo et al. [13] advanced these principles for linear classifiers, delivering practical algorithms with theoretical rigor.

Our work restricts the scope to the more mature unlearning area of classification tasks. Based on certified removal, we introduce feature decoupling and loss perturbation modules to enable a good approximation of retraining prediction performance after sample removal through localized updates.

## 3 Factor Decorrelation Enhanced Data Removal

In this section, we will detail the design of our DecoRemoval framework illustrated in Figure 1. DecoRemoval include two main modules: (1) Discriminative-preserving factor decorrelation by using random Fourier features to achieve spatial mapping and perform dimensionality reduction on input features (Section 3.2); (2) Smoothed data removal by integrating the random linear perturbation loss into unlearning training process to ensure privacy and security (Section 3.3). Moreover, we will integrate the core steps of 3.2 and 3.3 and introduce the main process of the DecoRemoval algorithm (Section 3.4). Next, we will explain them one by one.

### 3.1 Definitions

**Definition:** *Factor Decorrelation* [6, 12]: *Let $X \in \mathbb{R}^{n \times d}$ denote a dataset with $n$ samples and $d$ features, and let $\mathcal{A}$ be a learning algorithm trained on $X$. Factor Decorrelation refers to the process of reducing statistical dependencies (e.g., correlation) among features in $X$, with the objective of transforming it into a decorrelated representation $X'$ that preserves essential information for learning. As the correlations between features affect or even impair the model prediction, several works have focused on remove such correlation in the training process such as Random Fourier Features(RFF) [34]. RFF is used to approximate kernel functions and induce decorrelation by mapping the data to a higher-dimensional space. Given a kernel $k(x, y)$, RFF provides a feature*

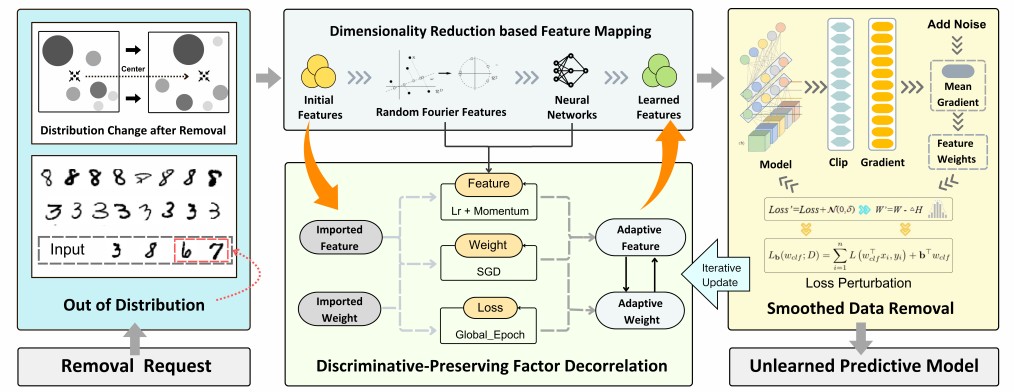

Figure 1: Factor decorrelation enhanced data removal. Overview of the DecoRemoval framework. It consists of two main modules: (1) Discriminative-Preserving Factor Decorrelation based on Random Fourier Features for spatial mapping and dimensionality reduction; (2) Smoothed Data removal through random linear perturbation loss integrated into the unlearning training process.

*mapping $\phi(x)$ such that the dot product $\langle\phi(x), \phi(y)\rangle \approx k(x, y)$. The transformation is given by:*

$$\phi(x) = \sqrt{\frac{2}{d}} \left[\cos(Wx + b)\right], \tag{1}$$

*where $W \in \mathbb{R}^{d \times d}$ is a random matrix, $b \in \mathbb{R}^d$ is a bias term, and $d$ is the number of random Fourier features. This transformation is designed to decorrelate the original features by embedding them in a higher-dimensional space.*

**Definition:** *Certified Removal [13]: Let $D$ be a training dataset and $A$ be a learning algorithm trained on $D$. $Range(A)$ is the value range of $A$. A data-removal mechanism $M$ is applied to $A(D)$, and we say that the removal mechanism $M$ performs $\epsilon$-certified removal ($\epsilon$-CR) for learning algorithm $A$ if, for all $S \subseteq Range(A)$ and $x \in D$, the following condition holds:*

$$e^{-\epsilon} \le \frac{P(M(A(D), D, x) \in S)}{P(A(D \setminus x) \in S)} \le e^{\epsilon} \tag{2}$$

The definition ensures that removing a single data point $x$ from the dataset $D$ will not affect the model's predictions by more than an exponential factor of $\epsilon$, preserving the model's stability.

## 3.2 Discriminative-Preserving Factor Decorrelation

To deal with OOD, we propose a discriminative-preserving factor decorrelation module that integrates Random Fourier Features (RFF) [12] into the neural network architecture. This module aims to decorrelate input features while preserving their class-discriminative structure, thereby promoting stable and generalizable learning. Specifically, input features are first projected into a higher-dimensional randomized feature space via an RFF-based transformation. This mapping reduces redundancy and statistical dependency among features, resulting in a smoother optimization landscape for subsequent layers.

While RFF aids in feature decorrelation, it may also disperse discriminative information across dimensions. Applying standard dimensionality reduction or naive loss functions without adapting to the transformed structure risks misaligning gradient directions with the true task objective. Such mismatch can destabilize training and impair generalization. To address this, we explicitly balances feature decorrelation with discriminative preservation. By aligning the transformation process with task-aware loss design, our approach maintains effective learning dynamics and avoids representation collapse.

**Random Fourier Feature Mapping:** Let the input feature vector for the $i$-th sample be denoted by $X_i \in \mathbb{R}^{m_X}$. The goal is to map $X_i$ into a high-dimensional feature space using the Random Fourier

Feature transformation. This transformation is based on the Fourier transform of a kernel function and is defined as follows:

$$Z_i = \sqrt{2} \cdot \cos(\omega X_i + \phi), \quad \omega \sim \mathcal{N}(0, I), \quad \phi \sim \text{Uniform}(0, 2\pi), \tag{3}$$

where $\omega \in \mathbb{R}^{m_Z}$ is sampled from a standard normal distribution, and $\phi \in [0, 2\pi]$ is sampled uniformly. The resulting vector $Z_i \in \mathbb{R}^{m_Z}$ is the transformed feature for the $i$-th sample. By utilizing this RFF mapping, we can approximate the kernel function $k(X, X')$ in a feature space without directly computing it, enabling the use of linear models in a high-dimensional feature space.

**Feature Decorrelation via Sample Weighting:** To further eliminate the correlation between the transformed features, we employ a sample weighting strategy that minimizes the dependence between features. Let $Z_{:,i}$ and $Z_{:,j}$ represent the $i$-th and $j$-th feature vectors of the transformed input. The goal is to reduce the statistical dependence between all pairs of features in the transformed space.

To achieve this, we utilize hypothesis testing statistics based on the cross-covariance between random variables. Let us define the cross-covariance operator $\Sigma_{AB}$ between two random variables $A$ and $B$, with corresponding kernel functions $k_A$ and $k_B$, as follows:

$$\langle h_A, \Sigma_{AB} h_B \rangle = \mathbb{E}_{AB}[h_A(A) h_B(B)] - \mathbb{E}_A[h_A(A)] \mathbb{E}_B[h_B(B)], \tag{4}$$

where $h_A \in \mathcal{H}_A$ and $h_B \in \mathcal{H}_B$ are elements of the Reproducing Kernel Hilbert Spaces (RKHS) corresponding to the random variables $A$ and $B$. The independence of the random variables $A$ and $B$ is indicated by the condition:

$$\Sigma_{AB} = 0 \iff A \perp B. \tag{5}$$

In our case, we use the cross-covariance between the transformed features $Z_{:,i}$ and $Z_{:,j}$ to measure their dependence. The partial cross-covariance matrix $\hat{\Sigma}_{AB}$ can be estimated as follows:

$$\hat{\Sigma}_{AB} = \frac{1}{n-1} \sum_{i=1}^{n} \left[ \left( u(Z_i) - \frac{1}{n} \sum_{j=1}^{n} u(Z_j) \right)^T \cdot \left( v(Z_i) - \frac{1}{n} \sum_{j=1}^{n} v(Z_j) \right) \right], \tag{6}$$

where $u$ and $v$ are the RFF transformations applied to the features $Z_i$ and $Z_j$, respectively. The Frobenius norm of this matrix is used as a measure of the dependence between features:

$$I_{AB} = \|\hat{\Sigma}_{AB}\|_F^2. \tag{7}$$

To further reduce feature dependence, we apply sample weighting. Let $w_i$ denote the sample weight for the $i$-th sample. The weighted partial cross-covariance matrix is computed as:

$$\hat{\Sigma}_{AB;w} = \frac{1}{n-1} \sum_{i=1}^{n} \left[ \left( w_i u(Z_i) - \frac{1}{n} \sum_{j=1}^{n} w_j u(Z_j) \right)^T \cdot \left( w_i v(Z_i) - \frac{1}{n} \sum_{j=1}^{n} w_j v(Z_j) \right) \right]. \tag{8}$$

**Optimization of Sample Weights:** The optimal sample weights $w^*$ are determined by minimizing the total dependence between all pairs of features. The optimization problem for the weights is formulated as:

$$w^* = \arg \min_{w \in \Delta_n} \sum_{1 \leq i < j \leq m_Z} \|\hat{\Sigma}_{Z_{:,i} Z_{:,j};w}\|_F^2, \tag{9}$$

where $\Delta_n = \{w \in \mathbb{R}^{n+} \mid \sum_{i=1}^{n} w_i = n\}$ ensures that the sample weights are positive and sum to $n$.

### 3.3 Smoothed Data Removal

To enhance the data removal mechanism with generalization across distributions, we propose a smoothed data removal module based on random linear perturbations. Specifically, we inject randomized noise into the training loss, which obfuscates gradient signals associated with removed or irrelevant samples. This smoothing effect suppresses sharp updates caused by individual data points, minimizing their influence on model predictions. As a result, the model becomes less sensitive to the removed data, reducing the risk of information leakage while maintaining stable learning behavior.

**Loss Perturbation for Data Removal:** To ensure that the removal of data does not inadvertently leak information about the removed samples, we begin by applying a loss perturbation technique at the training stage. This involves perturbing the loss function by adding a random linear term:

$$L_{\mathbf{p}}(w_{clf}; D) = \sum_{i=1}^{n} L\left(w_{clf}^{\top} x_i, y_i\right) + \mathbf{b}^{\top} w_{clf} \tag{10}$$

where $w_{clf} \in \mathbb{R}^d$ denotes the weight vector of the linear classifier (distinct from the sample weights $w$ used for decorrelation), and $\mathbf{b} \in \mathbb{R}^d$ is a random vector sampled from a prescribed distribution (e.g., Gaussian or uniform). The addition of $\mathbf{b}^{\top} w_{clf}$ serves to inject controlled stochasticity into the optimization process, thereby obscuring potential gradient signals associated with specific training instances. This perturbation mitigates the risk of overfitting and strengthens the model's robustness to sample removal under removal guarantees.

**Linear Authentication Removal:** After the loss perturbation, we proceed to the linear authentication removal step. To perform linear authentication removal, the deep learning network is split into two parts: the feature extraction layer parameters $w_{extr}$ and the linear classification layer parameters $w_{clf}$. This separation allows us to rewrite the loss function in terms of the linear classifier:

$$L(w_{clf}; D) = \sum_{i=0}^{n} L\left(w_{clf}^{\top} f(w_{extr}; f(w^0; x_i)), y_i\right) \tag{11}$$

where $w_{clf}^* = A(D) = \arg\min_{w_{clf}} L(w_{clf}; D)$. We assume that we aim to remove the last training sample $(x_n, y_n)$ from the dataset $D$, resulting in a modified dataset $D' = D \setminus (x_n, y_n)$.

To remove the sample $(x_n, y_n)$, we first compute the gradient of the loss function at $(x_n, y_n)$ and the Hessian of $L(\cdot; D')$ at $w_{clf}^*$:

$$\Delta = \nabla L(w_{clf}; (x_n, y_n)) \quad H_{w_{clf}^*} = \nabla^2 L(w_{clf}^*; D') \tag{12}$$

We then apply the Newton update removal mechanism $M$ as follows:

$$w_{clf}^- = M(w_{clf}^*, D, (x_n, y_n)) := w_{clf}^* + H_{w_{clf}^*}^{-1} \Delta \tag{13}$$

This update $H_{w_{clf}^*}^{-1} \Delta$ represents the influence function of the removed training sample on the vector $w_{clf}^*$. The training process of our DecorRemoval is described in Appendix.A.

**Robustness of Removal Under Perturbation.** To ensure the proposed removal mechanism remains valid under the perturbed loss, we analyze its impact on the gradient and Hessian. The perturbed loss is given by:

$$L_{\mathbf{p}}(w_{clf}; D) = \sum_{i=1}^{n} L(w_{clf}^{\top} x_i, y_i) + \mathbf{b}^{\top} w_{clf}, \tag{14}$$

where $\mathbf{b} \in \mathbb{R}^d$ is a random vector independent of individual samples. This linear term introduces a constant shift in the gradient:

$$\nabla L_{\mathbf{p}}(w_{clf}) = \nabla L(w_{clf}) + \mathbf{b}, \tag{15}$$

but leaves the Hessian unchanged:

$$\nabla^2 L_{\mathbf{p}}(w_{clf}) = \nabla^2 L(w_{clf}). \tag{16}$$

As the removal update depends on the Hessian and the gradient of the sample to be removed, which are both unaffected by $\mathbf{b}$, the update

$$w_{clf}^- = w_{clf}^* + H^{-1} \nabla L(w_{clf}^*; (x_n, y_n)) \tag{17}$$

remains valid. Hence, our removal mechanism is robust to the proposed linear perturbation, and the specific proof process is detailed in Appendix B.

# 4 Experiments

## 4.1 Datasets and Evaluation Metrics

Our approach are evaluated on five widely used datasets spanning multiple data modalities, including image, text, and structured features. Following the setup in Certified Removal [13], we consider three public datasets for classification tasks: MNIST [35], CIFAR-10 [36], and SST-2 [37]. MNIST consists of grayscale images of handwritten digits, where digits 3 and 8 are selected as in-distribution classes and the remaining digits are treated as out-of-distribution (OOD) data. CIFAR-10 contains 60,000 color images (32×32) evenly distributed across 10 object categories and serves as a standard benchmark for evaluating image classification methods. SST-2 is a binary sentiment analysis dataset derived from movie reviews, commonly used in text classification and language understanding tasks.

To assess the algorithm's applicability to privacy-sensitive structured data, we also include two social survey datasets: the 2015 China General Social Survey (CGSS) and the 2018 European Social Survey (ESS). Both datasets contain multi-label annotations related to self-reported happiness levels on a five-point ordinal scale [38, 39].

In the context of unlearning, it is essential to evaluate performance from three key aspects: utility, efficiency, and privacy protection. Given that certified unlearning methods provide theoretical guarantees on privacy, our experiments primarily focus on reporting utility and efficiency. We report accuracy (ACC) and weighted average F1 score as evaluation metrics to capture both overall classification performance and class imbalance sensitivity across diverse data types.

## 4.2 Experimental Setting

Following certified removal, we split the dataset into training, validation, and testing sets with a ratio of 7:1:1. Both training and validation sets consist solely of correctly labeled data to ensure standard supervised learning conditions. We train models independently on each category (e.g., training with data from class A), and then sample 10% of instances from other categories (e.g., class B). For testing, if these testy samples are predicted as "Not A", it means the unlearned model is correct. This creates an extreme testing scenario where semantic content mismatches the assigned label, enabling evaluation of the model's robustness against mislabeling or conceptual confusion.

In the experiment, the data unlearning baselines include Certified Removal(CR) [13], SISA (5 shards) [3], DP-SGD [40], Certified Unlearning (CU) [26]and SSD [27]. Both CR and DP-SGD provide strong theoretical guarantees under the framework of differential privacy. CU offers model-agnostic approximate unlearning strategies with soundness certificates and can be extended to deep neural networks [26]. SSD leverages the Fisher information matrix for parameter updates and represents the current state-of-the-art in certified removal [27]. In this paper, the source code provided by the baselines is used to fine-tune the parameters and obtain the optimal values to represent the experimental results of the baselines. For baselines which source code was not provided, this study reproduced the model design based on PyTorch framework. All methods set MLP as the backbone for all deep prediction models, ans are trained using a batch size of $d = 50$ and a total of $T = 20$ training epochs. For fair comparison, we fix the standard deviation parameters in DecoRemoval to $\delta = 10^{-3}$ and $std = 1$, and use a consistent optimization schedule with $num_s teps = 100$ across all experiments. Under these conditions, we subsequently applied the removal mechanism to each group separately. The related experiments in this study are conducted on four NVIDIA 4090D GPUs.

The time consumption for our DecoRemoval algorithm (DR) to remove data mainly comes from the computation of adaptive weights and the update operation of different features. We evaluate the performance of DecoRemoval algorithm across multiple datasets and varying data remove scales (1000, 3000, and 10000 samples). The full retraining from scratch (Retrain) is treated as the upper bound for accuracy, while Certified Removal serves as the baseline for efficiency comparison [13]. Our main codes and datasets are available at https://github.com/WUT-IDEA.

## 4.3 Unlearning Performance

Evaluated on five diverse datasets, DecoRemoval consistently achieves near-retraining performance in both accuracy and F1 score across varying removal scales. Unlike prior methods that focus primarily on privacy, our approach addresses feature correlation shifts via spatial mapping and randomized loss

Table 1: Comparison of ACC (%) and F1 scores across different methods and removed sample sizes (The closer to Retrain, the better)

| Dataset | Samples | Retrain | | CR [13] | | SISA [3] | | DP-SGD [40] | | SSD [27] | | CU [26] | | DR (Ours) | |
|---|---|---|---|---|---|---|---|---|---|---|---|---|---|---|---|
| | | ACC | F1 | ACC | F1 | ACC | F1 | ACC | F1 | ACC | F1 | ACC | F1 | ACC | F1 |
| MNIST | 1000 | 51.753 | 0.505 | 43.132 | 0.394 | 43.635 | 0.405 | 45.783 | 0.440 | 45.452 | 0.458 | 47.345 | 0.458 | **48.973** | **0.482** |
| | 3000 | 51.351 | 0.498 | 42.213 | 0.391 | 43.455 | 0.401 | 45.342 | 0.438 | 45.241 | 0.455 | 46.943 | 0.455 | **48.653** | **0.478** |
| | 10000 | 51.016 | 0.495 | 41.872 | 0.390 | 42.955 | 0.398 | 44.873 | 0.432 | 45.031 | 0.450 | 46.532 | 0.450 | **48.340** | **0.473** |
| CIFAR-10 | 1000 | 50.762 | 0.501 | 43.086 | 0.392 | 43.214 | 0.401 | 45.301 | 0.436 | 45.062 | 0.452 | 46.842 | 0.452 | **48.563** | **0.478** |
| | 3000 | 50.459 | 0.496 | 42.293 | 0.391 | 42.942 | 0.396 | 44.839 | 0.433 | 44.723 | 0.448 | 46.521 | 0.449 | **48.141** | **0.473** |
| | 10000 | 50.011 | 0.491 | 41.763 | 0.389 | 42.512 | 0.392 | 44.371 | 0.428 | 44.513 | 0.443 | 46.106 | 0.444 | **47.832** | **0.469** |
| SST-2 | 1000 | 91.764 | 0.843 | 89.705 | 0.808 | 89.975 | 0.817 | 90.452 | 0.825 | 89.983 | 0.818 | 89.942 | 0.813 | **90.451** | **0.827** |
| | 3000 | 91.545 | 0.840 | 89.651 | 0.801 | 89.760 | 0.814 | 90.356 | 0.820 | 89.865 | 0.816 | 89.765 | 0.808 | **90.387** | **0.825** |
| | 10000 | 91.142 | 0.839 | 89.478 | 0.796 | 89.653 | 0.809 | 90.101 | 0.816 | 89.873 | 0.816 | 89.673 | 0.805 | **90.375** | **0.821** |
| ESS | 1000 | 55.432 | 0.540 | 48.608 | 0.410 | 48.635 | 0.420 | 50.473 | 0.450 | 50.147 | 0.486 | 51.341 | 0.490 | **54.973** | **0.520** |
| | 3000 | 55.351 | 0.540 | 48.412 | 0.400 | 48.455 | 0.410 | 50.012 | 0.440 | 50.007 | 0.479 | 51.153 | 0.480 | **54.852** | **0.510** |
| | 10000 | 55.236 | 0.530 | 48.402 | 0.390 | 48.435 | 0.410 | 49.673 | 0.430 | 49.186 | 0.478 | 50.871 | 0.470 | **54.640** | **0.510** |
| CGSS | 1000 | 51.602 | 0.515 | 41.239 | 0.465 | 43.516 | 0.472 | 46.756 | 0.487 | 47.765 | 0.475 | 48.765 | 0.485 | **50.824** | **0.501** |
| | 3000 | 51.324 | 0.506 | 40.738 | 0.458 | 43.016 | 0.469 | 46.252 | 0.482 | 47.313 | 0.474 | 48.210 | 0.480 | **50.482** | **0.496** |
| | 10000 | 50.975 | 0.498 | 40.145 | 0.434 | 42.745 | 0.465 | 45.954 | 0.473 | 47.152 | 0.471 | 47.851 | 0.477 | **50.104** | **0.495** |

perturbation, ensuring both utility and robustness. Compared to recent baselines such as SSD and certified unlearning, DecoRemoval demonstrates stronger generalization, especially under large-scale deletions and noisy, high-dimensional data.

**DecoRemoval achieved the best performance in out of distribution scenarios across five datasets.** As shown in Table  ref Table: merged, DecoRemoval consistently achieved near-retraining accuracy and F1 score across five datasets and removal sizes, surpassing all existing Sota data removal models under out-of-distribution settings. On the ESS and CGSS datasets, which feature noisy and highly correlated survey data, our method achieves 54.9% and 50.8% accuracy after removing 1000 samples, with minimal degradation compared to full retraining (55.4% and 51.6%). In SST-2, DecoRemoval maintains over 90.3% accuracy across all remove scales, outperforming DecoRemoval by approximately 1 percentage point on both ACC and F1 metrics. Especially in image dataset scenes, DecoRemoval achieved significant improvement under out of distribution settings. This scenario is the main research and application goal of the current data removal mechanism.

**Compared to traditional data removal models that mainly focus on privacy processing, DecoRemoval performs better.** The biggest problem with data removal under out of distribution settings is the correlation between features that affects the distribution of data. The existing traditional data removal models mainly focus on privacy protection issues during the data removal process, using methods such as differential privacy to ensure model stability and security. However, when there are out of distribution changes in the data scene, these methods lack processing of the correlation between features, resulting in poor model performance. DecoRemoval identified the complexity of features in this scenario, achieved feature dimensionality reduction through spatial mapping, and ensured the privacy of the removal process by adding random loss perturbations, thus achieving optimal performance in ACC and F1 scores in OOD scenarios.

**Greater robustness compared to SOTA.** Compared with the latest methods such as SSD updated with Fisher matrix and optimized Certified Unlearning, DecoRemoval utilizes the advantage of spatial dimensionality reduction in feature correlation processing and exhibits stronger robustness in large-scale data deletion. For example, on the CGSS dataset with 10000 removed samples, our method achieved an F1 score of 0.495, while Zhang and Foster's methods only scored 0.477 and 0.471, respectively. This pattern is applicable to all datasets, indicating that our method has stronger generalization ability and stability.

## 4.4 Efficiency Analysis

In addition to unlearning fidelity, computational efficiency is a critical factor for practical deployment. While full retraining (Retrain) achieves optimal activity performance, it requires model reinitialization

Table 2: Comparison of running Time(s) for different removal methods (the closer to Certified Removal, the better)

| Samples | Dataset | Retrain | CR [13] | SISA [3] | DP-SGD [40] | CU [26] | SSD [27] | DR |
|---|---|---|---|---|---|---|---|---|
| 1000 | MNIST | 4643.100 | **21.312** | 1923.430 | 38.710 | 33.134 | 32.541 | 30.430 |
| | SST-2 | 61.500 | **0.074** | 24.670 | 0.124 | 0.105 | 0.095 | 0.097 |
| | ESS | 1539.100 | **7.420** | 648.400 | 12.800 | 12.458 | 12.127 | 11.500 |
| | CGSS | 615.690 | **8.450** | 362.400 | 15.346 | 14.541 | 14.377 | 14.353 |
| 3000 | MNIST | 11432.500 | **62.425** | 4321.200 | 102.510 | 91.329 | 90.786 | 81.420 |
| | SST-2 | 178.100 | **0.204** | 78.430 | 0.401 | 0.325 | 0.309 | 0.315 |
| | ESS | 3142.200 | **22.120** | 1448.200 | 40.500 | 38.718 | 37.812 | 35.200 |
| | CGSS | 1923.500 | **25.630** | 983.200 | 47.545 | 45.341 | 44.749 | 43.345 |
| 10000 | MNIST | 43123.500 | **190.342** | 13214.400 | 310.120 | 287.490 | 276.710 | 268.420 |
| | SST-2 | 598.400 | **0.715** | 236.400 | 1.030 | 0.904 | 0.891 | 0.894 |
| | ESS | 14532.500 | **78.400** | 6534.400 | 131.100 | 127.760 | 123.710 | 121.400 |
| | CGSS | 5893.400 | **81.423** | 3418.900 | 164.600 | 158.860 | 147.230 | 141.700 |

and complete retraining on the original dataset, making it computationally impractical for frequent or large-scale data removal scenarios. In contrast, DecoRemoval delivers strong unlearning performance at significantly lower computational cost. Unlike Retrain, which revisits the entire training set, DecoRemoval performs a lightweight fine-tuning procedure that specifically targets the removal-induced dominant feature directions, enabling it to efficiently handle removal scales ranging from 1,000 to 10,000 samples without full model retraining. Furthermore, compared to existing methods, DecoRemoval achieves better performance with lower overhead. Certified Removal and SISA rely on ensemble models or shard-based training pipelines, which incur significant complexity and computational burden [3, 13]. While DP-SGD offers built-in privacy guarantees, it injects substantial noise during training, resulting in lower post-removal accuracy despite its relative efficiency [40]. Finally, DecoRemoval strikes a more favorable balance between efficiency and unlearning quality than recent approaches proposed by Certified Unlearning (2024) and SSD (2024) [26, 27]. Across all settings, it consistently maintains performance close to the retraining upper bound while requiring far fewer computational resources, making it particularly suitable for real-world, large-scale, or streaming environments where fast and effective unlearning is essential.

## 4.5 Empirical Evaluation of Privacy Protection via Membership Inference Attacks

To further validate the privacy-preserving capabilities of DecoRemoval in the context of compliant machine learning (e.g., GDPR, CCPA), we evaluate its robustness against *Membership Inference Attacks* (MIAs)—a canonical threat model where an adversary attempts to determine whether a specific data sample was used in model training. Strong resistance to MIAs indicates effective data removal and reduced risk of information leakage.

Our evaluation is conducted under a realistic forgetting scenario with a privacy budget of $\epsilon = 1$. We compare DecoRemoval against several baselines. We assess MIA success rates across three model architectures (MLP, LSTM, Transformer) and two real-world datasets (ESS, CGSS), while also reporting accuracy and F1-score to evaluate utility preservation.

The results are summarized in Table 3. Key observations include:

- DecoRemoval consistently achieves lower MIA success rates than No-DP, demonstrating effective privacy protection after data removal.

- It outperforms SSD and CR across most settings, especially on Transformers, with better utility.

- On CGSS with LSTM, it achieves the lowest attack rate (50.2%) while maintaining higher accuracy than CR.

- Compared to DP-SGD, DecoRemoval offers significantly better utility at comparable privacy levels.

Table 3: Performance comparison of DecoRemoval and baselines on ESS and CGSS datasets across different architectures. Metrics: Accuracy (%), F1-score (%), and MIA success rate (%). Lower attack rate is better.

| Backbone | Method | ESS ACC ↑ | ESS F1 ↑ | ESS Attack ↓ | CGSS ACC ↑ | CGSS F1 ↑ | CGSS Attack ↓ |
|---|---|---|---|---|---|---|---|
| MLP | Retrain | **63.9** | **62.5** | 66.1 | **60.4** | **48.2** | 61.4 |
| | DP-SGD | 56.8 | 54.4 | 56.8 | 51.7 | 43.5 | **51.8** |
| | SSD | 59.0 | 56.9 | 57.3 | 53.2 | 45.2 | 52.5 |
| | CR | 60.7 | 59.8 | **56.7** | 55.6 | 45.9 | 52.4 |
| | DecoRemoval | 61.5 | 59.6 | 56.8 | 56.3 | 45.7 | 52.1 |
| LSTM | Retrain | **65.8** | **65.2** | 65.3 | **62.5** | **56.4** | 59.8 |
| | DP-SGD | 52.5 | 48.5 | 56.5 | 52.1 | 49.1 | 51.6 |
| | SSD | 55.6 | 53.2 | 56.8 | 53.8 | 53.2 | 50.9 |
| | CR | 57.1 | 53.5 | **55.9** | 55.4 | 54.1 | 50.4 |
| | DecoRemoval | 58.9 | 55.4 | 56.7 | 56.8 | 55.1 | **50.2** |
| Transformer | Retrain | **65.6** | **65.0** | 69.2 | **61.5** | **55.6** | 63.3 |
| | DP-SGD | 54.1 | 52.5 | 62.0 | 53.9 | 51.6 | 59.1 |
| | SSD | 57.3 | 55.8 | 62.3 | 54.8 | 52.7 | 58.8 |
| | CR | 57.7 | 56.1 | **60.5** | 55.4 | 52.9 | **57.4** |
| | DecoRemoval | 58.4 | 56.3 | 60.9 | 56.2 | 52.8 | 58.1 |

These results confirm that DecoRemoval not only stabilizes model distribution after data removal but also provides strong empirical privacy guarantees, making it suitable in privacy-sensitive applications.

### 4.6 Key Parameter Study

In Figure 2, we map the MINST dataset to a two-dimensional space for visualization after dimensionality reduction. It can be observed that after data removal, the distribution of the dataset and the position of the center point have undergone significant changes, which is consistent with our initial hypothesis about the impact of data removal. Appendix C shows DecoRemoval's performance depends on hidden layer size and RFF dimension, with optimal capacity ensuring good generalization.

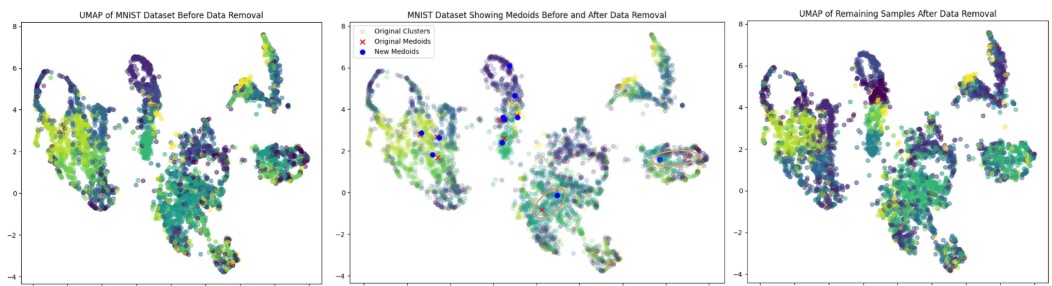

Figure 2: Results of MNIST dataset showing medoids before and after data removal

## 5 Conclusions and Future work

To tackle privacy preservation and out-of-distribution (OOD) challenges in predictive tasks, we integrate discriminative-preserving factor decorrelation with smoothed data removal. Our DecoRemoval mechanism enables efficient data unlearning while maintaining compliance with privacy regulations. The proposed method alleviates accuracy degradation commonly seen in traditional unlearning approaches under distribution shifts. Empirical results confirm its effectiveness and robustness in OOD scenarios. Future work will focus on applying this algorithm to various application domains that require robust machine unlearning capabilities, and how to work with current popular large models.

## 6 Acknowledge

This work is partially supported by NSFC, China (No.62276196). The authors also gratefully appreciate the anonymous reviewers for their valuable comments and constructive suggestions that greatly helped to improve the quality of the paper.

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

# A   DecoRemoval Training Algorithm

In 3.1 and 3.2, we have introduced feature dimensionality reduction methods using kernel based mapping and certified removal mechanism incorporating random loss perturbations. By utilizing the characteristics of random Fourier transform and sample feature weightings are fused through iterative loss fusion, and feature weights and related parameters are updated.So we summarized the overall algorithm flow of DecoRemoval and introduced the data removal mechanism after adding feature dimensionality reduction.

The DecoRemoval Algorithm aims to minimize feature correlation through the use of Random Fourier Features (RFF) and optimize sample weights efficiently. In the first step, the algorithm prepares the training dataset $D = \{(x_1, y_1), \ldots, (x_n, y_n)\}$ and initializes the neural network's feature extraction weights $w_{extr}$ and classification layer weights $w_{clf}$. The RFF transformation is applied to map the input features into a higher-dimensional space, resulting in transformed feature vectors $Z_i$. In the second step,the algorithm calculates the feature dependence between pairs of transformed features and optimizes sample weights $w_i$ to minimize this dependence, achieving decorrelation. For certified data removal in the third step, the influence of each sample is removed by applying a Newton update rule to the classifier weights. To prevent leakage of information from the removed samples, a perturbation term is added to the loss function. The final output includes the optimized classifier weights $w_{clf}^*$ and the transformed feature vectors $Z_i$.

---

**Algorithm 1:** Factor decorrelation enhanced data removal

---

**Inputs:**  training dataset $D = \{(x_1, y_1), \ldots, (x_n, y_n)\}$, kernel function $k$, neural network with feature
extraction layer $w_{extr}$, and linear classification layer $w_{clf}$;
**Hyperparameters:**  number of features $m_Z$, number of samples $n$;
**Step 1: Random Fourier Feature Mapping**;
**for** $i = 1, \ldots, n$ **do**
    Sample $\omega_i \sim \mathcal{N}(0, I)$, $\phi_i \sim \text{Uniform}(0, 2\pi)$;
    $Z_i = \text{RFF\_transform}(X_i, \omega_i, \phi_i)$;

**Step 2: Discriminative-Preserving Factor Decorrelation**;
**for** *each pair* $(Z_i, Z_j)$ **do**
    Compute feature dependence: $I_{ij} = \text{FrobeniusNorm}(\hat{\Sigma}_{ij})$;

Apply sample weights $w_i$ to minimize feature dependence: $w^* = \text{Optimize Weights}(\hat{\Sigma}_{AB;w})$;
**Step 3: Smoothed Data Removal**;
**for** *each sample* $(x_n, y_n)$ **do**
    Compute gradient: $\Delta = \nabla L(w_{clf}; (x_n, y_n))$;
    Compute hessian: $H_{w_{clf}^*} = \nabla^2 L(w_{clf}^*; D')$;
    Update classifier: $w_{clf}^- = w_{clf}^* + H_{w_{clf}^*}^{-1} \Delta$;

Add random linear term to the loss: $L_{\mathbf{b}} = L + \mathbf{b}^\top w_{clf}$;
**Return:** Optimized classifier parameters $w_{clf}^*$ and transformed feature vectors $Z_i$.

---

# B   Proof of Robustness Under Loss Perturbation

In this section, we provide a formal proof that adding a linear perturbation term to the training loss does not affect the correctness of the linear authentication removal mechanism.

## B.1   Perturbed Loss Function

We consider a perturbed loss function of the following form:

$$L_{\mathbf{p}}(w_{clf}; D) = \sum_{i=1}^{n} L\left(w_{clf}^\top x_i, y_i\right) + \mathbf{b}^\top w_{clf}, \tag{18}$$

where $w_{clf} \in \mathbb{R}^d$ is the linear classifier, and $\mathbf{b} \in \mathbb{R}^d$ is a random vector sampled from a fixed distribution (e.g., Gaussian or uniform). The term $\mathbf{b}^\top w_{clf}$ introduces controlled randomness to the optimization process.

## B.2 Gradient and Hessian Analysis

Let $L_{\text{orig}}(w_{clf}) = \sum_{i=1}^{n} L(w_{clf}^{\top} x_i, y_i)$ denote the original loss function. Then, the gradient of the perturbed loss is:

$$\nabla L_{\mathbf{p}}(w_{clf}) = \nabla L_{\text{orig}}(w_{clf}) + \mathbf{b}. \qquad (19)$$

The Hessian of the perturbed loss is:

$$\nabla^2 L_{\mathbf{p}}(w_{clf}) = \nabla^2 L_{\text{orig}}(w_{clf}) + \nabla^2(\mathbf{b}^{\top} w_{clf}) = \nabla^2 L_{\text{orig}}(w_{clf}), \qquad (20)$$

since the second derivative of a linear term is zero. Thus, the curvature of the loss landscape (captured by the Hessian) remains unchanged by the perturbation.

## B.3 Effect on Removal Update

Assume that we wish to remove the final sample $(x_n, y_n)$ from the dataset $D$, yielding the modified dataset $D' = D \setminus (x_n, y_n)$. The Newton-based removal update is given by:

$$w_{clf}^{-} = w_{clf}^{*} + H^{-1} \nabla L(w_{clf}^{*}; (x_n, y_n)), \qquad (21)$$

where $w_{clf}^{*}$ is the minimizer of the loss (perturbed or unperturbed), and $H$ is the Hessian of the loss over $D'$ evaluated at $w_{clf}^{*}$.

Under the perturbed loss, we denote the minimizer as $w_{clf}^{*p}$, which satisfies:

$$\nabla L_{\text{orig}}(w_{clf}^{*p}) + \mathbf{b} = 0 \quad \Rightarrow \quad \nabla L_{\text{orig}}(w_{clf}^{*p}) = -\mathbf{b}. \qquad (22)$$

However, this constant offset in the gradient does not affect the sample-specific gradient $\nabla L(w_{clf}; (x_n, y_n))$, nor the Hessian $H$, since both are independent of $\mathbf{b}$. Therefore, the removal update remains:

$$w_{clf}^{-} = w_{clf}^{*p} + H^{-1} \nabla L(w_{clf}^{*p}; (x_n, y_n)), \qquad (23)$$

which is structurally identical to the original update formula. As a result, the removal mechanism is preserved under loss perturbation.

## B.4 Conclusion

The addition of a linear perturbation term does not interfere with the linear removal update. The gradient shift induced by $\mathbf{b}$ is constant and does not impact the relative influence of any individual data point. The Hessian remains unchanged, and the Newton update retains its validity. This confirms the robustness of our removal strategy under smoothed loss perturbations.

# C  In-distribution Setting Experiment

For the in distribution data scenario, we evaluated the accuracy and F1 score of Certified Removal(CR), DecoRemoval(DR), Certified Unlearning(CU), and SSD on different datasets. This setting enables us to systematically evaluate the robustness of the model to domain differences and its generalization ability when the training distribution remains largely unchanged.

Table 4: Comparison of ACC (%) and F1 scores across different methods under in-distribution setting

| Method | MNIST | | CIFAR10 | | SST-2 | | ESS | | CGSS | |
|---|---|---|---|---|---|---|---|---|---|---|
| | ACC | F1 | ACC | F1 | ACC | F1 | ACC | F1 | ACC | F1 |
| CR | 91.974 | 0.930 | 90.878 | 0.931 | 96.103 | 0.979 | 92.863 | 0.939 | 92.798 | 0.921 |
| SSD | 92.429 | 0.940 | 92.320 | 0.964 | 96.982 | 0.987 | 93.754 | 0.943 | 93.305 | 0.930 |
| CU | 94.213 | 0.964 | 93.271 | 0.946 | 96.923 | 0.982 | 93.671 | 0.950 | 93.193 | 0.945 |
| DR (Ours) | **95.841** | **0.968** | **95.034** | **0.963** | **97.011** | **0.988** | **93.852** | **0.969** | **93.860** | **0.948** |

# D   Unlearning Performance

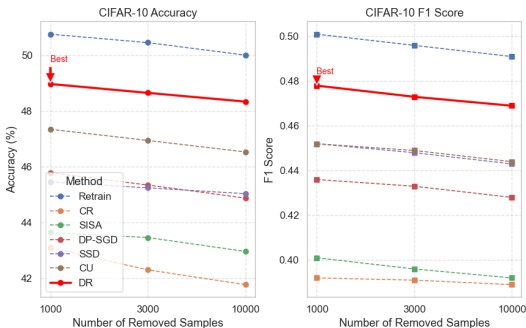

Figure 3:  Removal performance of different numbers of removed samples in CIFAR-10

**Accuracy and F1 Score.**
These results demonstrate the effectiveness of DecoRemoval, our factor decorrelation-based data removal strategy. By adaptively reweighting feature dimensions to suppress redundant correlations, DecoRemoval eliminates the need for full retraining, ensuring fast and efficient data removal. It performs well across various datasets and conditions, providing high forgetting fidelity and accuracy while balancing computational efficiency and rigorous unlearning, making it ideal for privacy-sensitive applications.

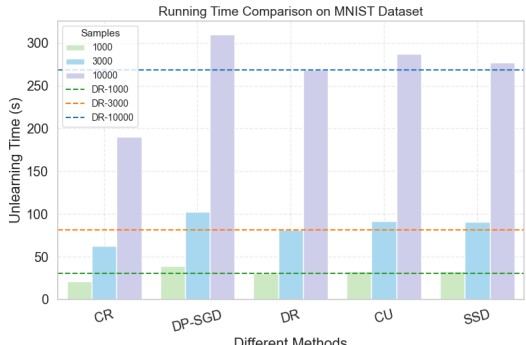

Figure 4:  Efficiency of different removed numbers in CIFAR-10 datasets

**Efficiency.**  Overall, our DecoRemoval achieves an effective balance between high forgetting fidelity and practical efficiency, especially when facing out of distribution situations as shown in Figure 4. It can achieve performance close to the level of retraining without incurring the huge cost of comprehensive retraining, and has a significant improvement in the balance between accuracy and efficiency compared to existing data removal mechanisms. This makes it a highly promising solution for scalable and trustworthy machine learning.

# E   Key Parameter Study

Our DecoRemoval algorithm adjusts the hidden layer dimension and the random Fourier transform dimension of the neural network under out-of-distribution settings, and the tested results are reported in terms of Accuracy and F1 score on the datasets, as shown in Figure 5. Specifically, when the hidden layer dimension is about 80, the accuracy and F1 score of the two happiness datasets ESS and CGSS, as well as the sentiment text dataset SST-2, reach their maximum values, outperforming other traditional models and approaching the results of retraining. For the image dataset MNIST, the performance reaches its best when the hidden layer dimension is about 400.

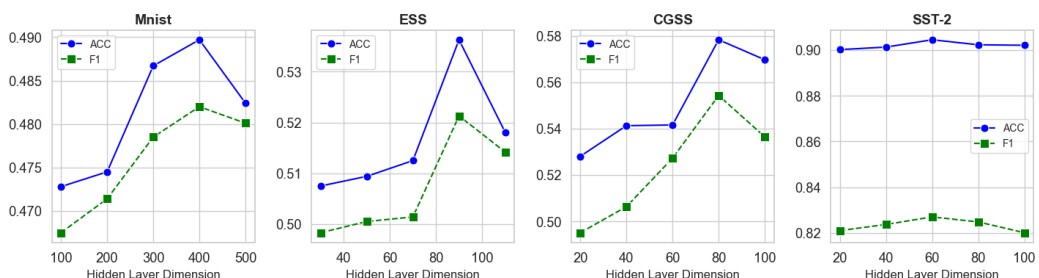

Figure 5:  Results of different hidden layer dimensions in adaptive weighted factor decorrelation

The results indicate that the DecoRemoval algorithm can significantly improve its generalization ability in the presence of out-of-distributed data.

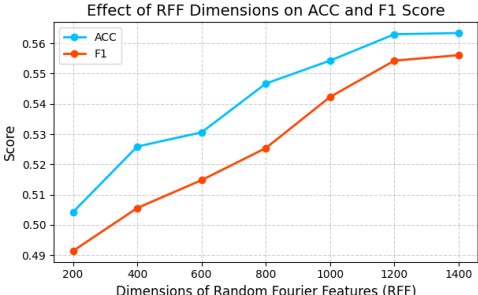

Figure 6: Experimental results of ESS in different RFF dimensions

**RFF dimensions**
During training with varying RFF dimensions, we observed that both model accuracy and F1 score consistently improved as the dimensionality increased, peaking at 1000 dimensions as shown in Figure 6. Beyond this point, performance gains plateaued, showing minimal change with further increases. These findings suggest that the DecoRemoval algorithm effectively enhances generalization under non-out-of-distribution conditions, particularly when equipped with a sufficiently expressive RFF representation.

## F Backbones

Table 5: Comparison of ESS Deletion Efficiency of Different Backbones in Deep Predictive Models (the closer to Retrain, the better)

| Backbones | Retrain | | Certified Removal | | FD-DR(ours) | |
|---|---|---|---|---|---|---|
| | Time(s) ↓ | ACC(%) ↑ | Time(s) ↓ | ACC(%) ↑ | Time(s) ↓ | ACC(%) ↑ |
| MLP | 1539.1 | 55.4 | 7.4 | 48.6 | 11.5 | **54.8** |
| LSTM | 3583.1 | 53.9 | 20.4 | 48.2 | 28.6 | **52.5** |
| Transformer | 1956.2 | 54.1 | 16.4 | 48.5 | 15.4 | **53.1** |

## G Limitations

**Limited Exploration Beyond Feature-Level Decorrelation.** This work primarily focuses on mitigating out-of-distribution (OOD) challenges through feature-level factor decorrelation. While effective, it leaves open how this approach interacts with other common OOD handling techniques such as data augmentation, adversarial training, or ensemble learning. A promising direction for future work is to explore how these methods can be systematically integrated with data removal strategies to enhance both generalization and unlearning robustness.

**Trade-off Between Certified Removal and Accuracy.** Although certified removal offers strong theoretical guarantees and is efficient for linear layers, it may be suboptimal in scenarios where high predictive accuracy is critical, such as medical diagnosis or financial forecasting. In such cases, privacy strength could be moderately relaxed in favor of more expressive unlearning methods, such as influence function-based unlearning or fine-tuning-based approximate unlearning, to strike a better balance between utility and privacy.

