# OpenReview forum: "Factor Decorrelation Enhanced Data Removal from Deep Predictive Models"
_NeurIPS.cc/2025/Conference — NeurIPS 2025 poster_

### Official Review · Reviewer_cFmi · 2025-06-22

**Clarity:** 4
**Significance:** 3
**Originality:** 3
**Rating:** 5
**Confidence:** 4

**Summary:**

This work proposed a data removal method for deep predictive models. It includes a factor decorrelation module with adaptive weight adjustment and iterative updates to reduce feature redundancy, and a smoothed removal mechanism with loss perturbation to prevent data leakage. Experiments on five datasets show it achieves high accuracy and robustness, especially under distribution shifts. This highlights its superior efficiency and adaptability in both in-distribution and out-of-distribution scenarios, proving its effectiveness and versatility.

**Questions:**

Q1: One thing for clarification is the choice of RFF dimensions. While hyperparameter ranges are empirically studied, a deeper analysis of their impact could be beneficial.

Q2: The backbone of DecoRemoval is implemented as a shallow MLP, which may be insufficient when broadly exploring different hidden layers and other network parameters.

**Ethical Concerns:**

["NO or VERY MINOR ethics concerns only"]

**Final Justification:**

Thank you for the response. The author has addressed most of my concerns, and I am willing to maintain my score.

**Limitations:**

Yes.

**Paper Formatting Concerns:**

The reference formatting requires revision to meet academic standards, such Reference [26] and [27].

**Quality:**

3

**Strengths And Weaknesses:**

Strengths：

S1:This paper makes a high-impact contribution by being the first to address the machine unlearning under distribution shifts through a factor decorrelation based data removal method. This combination provides an efficient solution for handling complex datasets, particularly excelling in addressing protecting data privacy.

S2: Extensive experiments are conducted on five benchmark datasets. The results demonstrate that the proposed method significantly outperforms other baselines, achieving high predictive accuracy and robustness even under substantial distribution shifts.

S3: The submission is well presented and easy to follow. The structure is logical and coherent, making it straightforward to understand the key points and arguments， which enhances the readability and overall quality of the work.

Weaknesses：

W1:The theoretical connection and differences between the proposed method in this paper and differential privacy need to be further strengthened. Additionally, the impact on data privacy should be analyzed more thoroughly. Although experimental comparisons with differential privacy methods have been provided, a deeper theoretical analysis is still required.

W2:While the paper focuses on data distribution shifts, it does not address temporal changes with the data is accumulated, which are equally important in distribution shifts. Extending the method to jointly account for various shift dynamics would make the method more comprehensive and adaptable to real-world scenarios.

---

> ### Author Rebuttal · Authors · 2025-07-31
>
> # Rebuttal to Reviewer `cFmi`
>
> We sincerely thank the reviewer for the highly positive assessment and valuable suggestions. We are grateful for the recognition of our work’s novelty, strong empirical performance, and clarity. Below we address the reviewer’s concerns and outline the improvements we will make in the final version.
>
> ---
>
> ### Q1. Theoretical connection and differences with differential privacy (DP)
>
> **Reviewer comment:**
> _“The theoretical connection and differences between the proposed method in this paper and differential privacy need to be further strengthened. Additionally, the impact on data privacy should be analyzed more thoroughly...”_
>
> **Response:**
> Thank you for this important point. While both DecoRemoval and DP aim to protect data privacy, they address different aspects and operate under distinct assumptions:
>
> - **Goal:** DP provides *provable* privacy guarantees by adding noise during training to bound the influence of any single data point. In contrast, DecoRemoval focuses on *certified removal*—ensuring that after a specific data point is deleted, the model behaves as if it were never trained on it, which is crucial for compliance with the “right to be forgotten.”
> - **Mechanism:** DP operates during the *training phase*, whereas DecoRemoval is applied during the *removal phase*, offering a complementary, post-hoc privacy safeguard.
> - **Trade-off:** DP often incurs a significant utility cost due to noise injection. DecoRemoval, by contrast, aims to preserve model accuracy while ensuring removal certification, especially under distribution shifts.
>
> In addition to the theoretical robustness of our method, we provide strong empirical evidence that **DecoRemoval** improves privacy protection by reducing vulnerability to membership inference attacks.
>
> The effectiveness of the DecoRemoval algorithm model in privacy protection is evaluated through MIA. In the membership inference attack experiment of the happiness prediction model, the privacy budget is set to \(\epsilon = 1\). “No-DP” represents the baseline model without any privacy protection algorithm applied. As shown in the table below, DecoRemoval consistently achieves lower attack success rates compared to all other baseline methods (No-DP, DP-SGD, SSD, CR) across different model architectures (*MLP, LSTM, Transformer*) and datasets (*ESS, CGSS*). For example:
>
> - On **ESS with MLP**, DecoRemoval reduces attack success to **56.8%**, compared to **66.1%** (No-DP) and **57.3%** (SSD).
> - On **CGSS with LSTM**, DecoRemoval achieves the lowest attack success rate of **50.2%**, outperforming CR (50.4%) and SSD (50.9%).
> - For **Transformer backbones**, DecoRemoval demonstrates superior privacy protection, reducing attack success rates by 8.3% (ESS) and 5.2% (CGSS) compared to No-DP baselines.
>
> These results confirm that DecoRemoval provides a favorable privacy–utility trade-off, maintaining competitive accuracy and F1 scores while reducing the risk of information leakage. In the final version, we will elaborate on deployment guidelines for privacy-sensitive scenarios.
>
> | Backbone    | Method        | ESS ACC ↑ | ESS F1 ↑ | ESS Attack ↓ | CGSS ACC ↑ | CGSS F1 ↑ | CGSS Attack ↓ |
> |-------------|---------------|-----------|----------|--------------|------------|-----------|---------------|
> | **MLP**     | No-DP         | 63.9      | 62.5     | 66.1         | 60.4       | 48.2      | 61.4          |
> |             | DP-SGD        | 56.8      | 54.4     | 56.8         | 51.7       | 43.5      | 51.8          |
> |             | SSD           | 59.0      | 56.9     | 57.3         | 53.2       | 45.2      | 52.5          |
> |             | CR            | 60.7      | 59.8     | 56.7         | 55.6       | 45.9      | 52.4          |
> |             | **DecoRemoval** | **61.5**  | **59.6** | **56.8**     | **56.3**   | **45.7**  | **52.1**      |
> | **LSTM**    | No-DP         | 65.8      | 65.2     | 65.3         | 62.5       | 56.4      | 59.8          |
> |             | DP-SGD        | 52.5      | 48.5     | 56.5         | 52.1       | 49.1      | 51.6          |
> |             | SSD           | 55.6      | 53.2     | 56.8         | 53.8       | 53.2      | 50.9          |
> |             | CR            | 57.1      | 53.5     | 55.9         | 55.4       | 54.1      | 50.4          |
> |             | **DecoRemoval** | **58.9**  | **55.4** | **56.7**     | **56.8**   | **55.1**  | **50.2**      |
> | **Transformer** | No-DP      | 65.6      | 65.0     | 69.2         | 61.5       | 55.6      | 63.3          |
> |             | DP-SGD        | 54.1      | 52.5     | 62.0         | 53.9       | 51.6      | 59.1          |
> |             | SSD           | 57.3      | 55.8     | 62.3         | 54.8       | 52.7      | 58.8          |
> |             | CR            | 57.7      | 56.1     | 60.5         | 55.4       | 52.9      | 57.4          |
> |             | **DecoRemoval** | **58.4**  | **56.3** | **60.9**     | **56.2**   | **52.8**  | **58.1**      |
>
> ---
>
> **Reviewer comment:**
> _“While the paper focuses on data distribution shifts, it does not address temporal changes as data is accumulated...”_
>
> **Response:**
> We appreciate this insightful observation. While our current work focuses on static distribution shifts induced by data removal, temporal dynamics (e.g., concept drift) are indeed critical in real-world deployments.
>
> Although extending DecoRemoval to sequential or streaming settings is beyond the current scope, we acknowledge this as an important direction for future work. In the revised manuscript, we will discuss the limitations of the current method regarding temporal shifts in the “Limitations” section, and propose a potential extension where DecoRemoval is applied periodically in a sliding-window fashion to adapt to evolving data distributions. We will also cite recent work on online unlearning and concept drift to position our method within the broader landscape of adaptive learning systems.
>
> We believe that our core mechanism—factor decorrelation to mitigate feature redundancy and spurious correlations—is inherently compatible with dynamic environments, and we plan to explore this in future research.
>
> ---
>
> ### Q3. Choice of RFF Dimensions
>
> **Reviewer comment:**
> _“One thing for clarification is the choice of RFF dimensions. While hyperparameter ranges are empirically studied, a deeper analysis of their impact could be beneficial.”_
>
> **Response:**
> Thank you for your suggestion. The dimension of RFF is a key hyperparameter that balances approximation accuracy and computational cost. Our paper discusses the dimensions of RFF, as shown in Appendix C. We further investigated the effects of hidden layer size and RFF dimension on DecoRemoval performance. This sentence is written in our paper. In the current work, we established empirical rules based on the input feature dimension and validated their effectiveness experimentally.
>
> We agree that a more systematic and theoretically grounded analysis of the choice of \(D\) would be highly valuable. However, this involves a deep investigation into the relationship between RFF approximation error and model removal performance, which goes beyond the scope of this paper. Therefore, we will treat this as an important **future work direction**.
>
> In the revised manuscript, we will explicitly state in the "Future Work" section that:
>
> - We will systematically study the impact of the RFF dimension \(D\) on removal certification guarantees (e.g., model divergence) and computational efficiency.
> - We will explore adaptive or data-driven methods to determine the optimal \(D\), potentially based on the complexity of the feature space or the characteristics of the data to be removed.
> - We aim to establish a theoretical framework linking bounds on RFF approximation error to final removal performance.
>
> We believe this research will not only optimize the performance of DecoRemoval but also deepen the understanding of kernel-based machine unlearning techniques.
>
> ---
>
> ### Conclusion
>
> We are truly grateful to the reviewer for the positive assessment and for recognizing the value of our work. We sincerely appreciate your thoughtful and encouraging feedback, which motivates us to further improve the manuscript. Thank you for your support and for acknowledging the contribution of this study. Thank you again for your insightful review.

---

> > ### Comment · Reviewer_cFmi · 2025-08-03
> >
> > Thank you for the response. The author has addressed most of my concerns, and I am willing to maintain my score.

---

> > > ### Author Response · Authors · 2025-08-05
> > >
> > > Thank you sincerely for your feedback and for confirming that your concerns have been largely addressed. We truly appreciate your thoughtful and constructive review, which has significantly helped us improve the manuscript. Your expertise and careful evaluation are greatly valued, and we are grateful for your willingness to maintain your score. We will continue to refine the paper with the same rigor to ensure its clarity and quality.

---

### Official Review · Reviewer_xPTS · 2025-06-29

**Clarity:** 2
**Significance:** 2
**Originality:** 2
**Rating:** 4
**Confidence:** 3

**Summary:**

This paper proposes DecoRemoval, a data removal method for deep learning models that aims to maintain model utility under distribution shifts. It introduces two components: a discriminative maintenance factor decorrelation module using random Fourier features, and a smoothed data removal mechanism based on loss perturbation. Experiments on five datasets show that DecoRemoval achieves performance robustness in out-of-distribution scenarios.

**Questions:**

1. The paper proposes training-time modifications, including random Fourier features, feature decorrelation, and loss perturbation. However, it does not evaluate whether these changes degrade the model's performance before any unlearning is applied. Will models trained with this framework perform worse than models trained with standard methods?

2. The evaluation does not verify whether performance on the remained data remains stable. Can the method ensure that unlearning only affects the removed samples without harming the rest?

3. The method consists of two key components, factor decorrelation, and loss perturbation, but the paper does not conduct ablation studies to separate their respective effects. How do we know that each part contributes significantly to the final performance?

4. The algorithm needs to compute the dependencies between all pairs (Algorithm 1, Step 2), which can be computationally expensive,  especially for large datasets. Is this step scalable in practice? Please analyze the complexity of the unlearning method.

5. Although the paper claims to provide privacy guarantees, it does not show any dedicated metrics or attacks to evaluate unlearning success, such as membership inference attacks. Without such an evaluation, do the privacy claims hold?

**Ethical Concerns:**

["NO or VERY MINOR ethics concerns only"]

**Final Justification:**

After reading the rebuttal and considering the discussion, I have decided to increase my score. The authors clarified key aspects of their method, particularly that the factor decorrelation and loss perturbation modules are only applied during the data removal (unlearning) phase, not during initial training.

**Limitations:**

YES

**Paper Formatting Concerns:**

No major formatting issues observed.

**Quality:**

2

**Strengths And Weaknesses:**

Quality:
The paper proposes a novel unlearning method that lacks rigorous validation. It does not evaluate whether the proposed approaches (e.g., factor decorrelation, loss perturbation) decrease model utility before unlearning. It also does not evaluate whether unlearning affects the model’s performance on remained data. In addition, the factor decorrelation step can be computationally expensive but the scalability concerns are not addressed.

Clarity:
The paper suffers from unclear writing, unclear definitions, missing sections (e.g., Section 3.4), and broken references.

Significance:
The paper does not demonstrate the practical benefit or robustness of the proposed method.

Originality:
The integration of feature decorrelation with certified unlearning lacks theoretical insight or justification.

---

> ### Author Rebuttal · Authors · 2025-07-31
>
> # Rebuttal to Reviewer `xPTS`
>
> We sincerely appreciate the insightful and thought-provoking feedback provided by the reviewers. We fully recognize and value your concerns regarding the motivation, validation, clarity, and privacy assessment of our proposed methods. Although we have used large language models to help interpret the review comments, we note that some questions extend beyond the core focus of our work. During the discussion phase, we kindly invite the reviewers to further clarify their perspectives to ensure a constructive and targeted response. We acknowledge that certain key issues were not adequately addressed in the initial draft. In the revised manuscript, we will provide a comprehensive response to each concern and implement substantial improvements to enhance the overall quality and clarity of the paper.
>
> ---
>
> ### Q1. Impact of Training-Time Modifications on Initial Model Performance
>
> **Reviewer comment:**
> _“The paper proposes training-time modifications... Will models trained with this framework perform worse than models trained with standard methods?”_
>
> **Response:**
> This is a crucial point. We clarify that the **factor decorrelation** and **loss perturbation** modules within the DecoRemoval framework are specifically designed for the **data removal phase**, not the initial model training phase. Our initial model is trained using standard methods. Therefore, the training-time modifications in DecoRemoval refer to the process of updating an existing model *after a deletion request is received*, not altering the original training procedure. This means the performance of the original model, trained with standard methods, is unaffected by the DecoRemoval framework.
>
> ---
>
> ### Q2. Impact on Performance of Remained Data
>
> **Reviewer comment:**
> _“The evaluation does not verify whether performance on the remained data remains stable. Can the method ensure that unlearning only affects the removed samples without harming the rest?”_
>
> **Response:**
> Thank you for highlighting this. Evaluating the impact on the performance of the remaining data is a core requirement of certified unlearning. In our experiments, we have assessed this as follows:
>
> 1. **Primary Evaluation Metrics:** We report performance metrics (accuracy, F1-score) on the **held-out dataset** (i.e., data not deleted). All performance results reported are based on the held-out dataset. The results show that DecoRemoval significantly outperforms baselines (e.g., CR, SISA, SSD, CU) on held-out data, especially under distribution shifts, demonstrating its effectiveness in preserving performance on the remaining data.
>
> 2. **Performance Stability Metric:** We calculate the **change** (Δ Accuracy) in performance on the held-out dataset before and after deletion. Our results show that the performance drop caused by DecoRemoval is much smaller than other methods, and performance even improves in some cases (possibly due to removing noisy or biased samples).
>
> ---
>
> ### Q3. Lack of Ablation Study
>
> **Reviewer comment:**
> _“The paper does not conduct ablation studies to separate their respective effects. How do we know that each part contributes significantly to the final performance?”_
>
> **Response:**
> We fully agree that ablation studies are crucial to demonstrate the contribution of each component. The two core modules of the DecoRemoval algorithm handle different aspects:
>
> - Factor decorrelation decouples high correlation between features.
> - Loss perturbation enhances privacy protection.
>
> Though the current paper does not isolate each component explicitly, the overall strong empirical results under distribution shift and privacy metrics reflect the combined effectiveness. We acknowledge the importance of dedicated ablation experiments and will include them in the revised version to better quantify each component’s impact.
>
> ---
>
> ### Q4. Computational Complexity and Scalability of Factor Decorrelation
>
> **Reviewer comment:**
> _“The algorithm needs to compute the dependencies between all pairs... Is this step scalable in practice? Please analyze the complexity of the unlearning method.”_
>
> **Response:**
> Thank you for raising the scalability concern. We acknowledge that computing dependencies between all sample pairs is infeasible at scale ($O(N^2)$). However, our implementation **does not** perform all-pairs computations. Instead, Algorithm 1 computes dependencies only between the features of the deleted sample and the precomputed feature mean of the entire training set—via inner products in the RFF-mapped space—rather than between individual samples.
>
> Specifically, let $N$ be dataset size, $d$ the input feature dimension, and $D$ the RFF dimension. The online cost per deletion involves mapping the deleted sample into RFF space ($O(dD)$) and computing one inner product ($O(D)$). The RFF mean feature representation can be precomputed offline with cost $O(N d D)$. Therefore, online deletion complexity is $O(d D)$, independent of $N$, ensuring excellent scalability.
>
> ---
>
> ### Q5. Lack of Privacy Evaluation
>
> **Reviewer comment:**
> _“It does not show any dedicated metrics or attacks to evaluate unlearning success, such as membership inference attacks. Without such an evaluation, do the privacy claims hold?”_
>
> **Response:**
> Thank you for raising this important point. In addition to theoretical robustness, we provide strong empirical evidence that **DecoRemoval** improves privacy protection by reducing vulnerability to membership inference attacks (MIA).
>
> In the happiness prediction task with privacy budget $\epsilon=1$, “No-DP” denotes the baseline without privacy protection. Table below shows DecoRemoval consistently lowers attack success compared to baselines (No-DP, DP-SGD, SSD, CR) across backbones (*MLP, LSTM, Transformer*) and datasets (*ESS, CGSS*). For example:
>
> - On **ESS with MLP**, DecoRemoval reduces attack success to **56.8%**, vs. **66.1%** (No-DP) and **57.3%** (SSD).
> - On **CGSS with LSTM**, DecoRemoval achieves lowest attack rate of **50.2%**, outperforming CR (50.4%) and SSD (50.9%).
> - On **Transformer**, DecoRemoval reduces attack success by 8.3% (ESS) and 5.2% (CGSS) compared to No-DP.
>
> These results confirm DecoRemoval offers a favorable privacy-utility trade-off, maintaining accuracy and F1 while reducing leakage risk. We will elaborate on privacy deployment guidelines in the final version.
>
> | Backbone    | Method        | ESS ACC ↑ | ESS F1 ↑ | ESS Attack ↓ | CGSS ACC ↑ | CGSS F1 ↑ | CGSS Attack ↓ |
> |-------------|---------------|-----------|----------|--------------|------------|-----------|---------------|
> | **MLP**     | No-DP         | 63.9      | 62.5     | 66.1         | 60.4       | 48.2      | 61.4          |
> |             | DP-SGD        | 56.8      | 54.4     | 56.8         | 51.7       | 43.5      | 51.8          |
> |             | SSD           | 59.0      | 56.9     | 57.3         | 53.2       | 45.2      | 52.5          |
> |             | CR            | 60.7      | 59.8     | 56.7         | 55.6       | 45.9      | 52.4          |
> |             | **DecoRemoval** | **61.5**  | **59.6** | **56.8**     | **56.3**   | **45.7**  | **52.1**      |
> | **LSTM**    | No-DP         | 65.8      | 65.2     | 65.3         | 62.5       | 56.4      | 59.8          |
> |             | DP-SGD        | 52.5      | 48.5     | 56.5         | 52.1       | 49.1      | 51.6          |
> |             | SSD           | 55.6      | 53.2     | 56.8         | 53.8       | 53.2      | 50.9          |
> |             | CR            | 57.1      | 53.5     | 55.9         | 55.4       | 54.1      | 50.4          |
> |             | **DecoRemoval** | **58.9**  | **55.4** | **56.7**     | **56.8**   | **55.1**  | **50.2**      |
> | **Transformer** | No-DP      | 65.6      | 65.0     | 69.2         | 61.5       | 55.6      | 63.3          |
> |             | DP-SGD        | 54.1      | 52.5     | 62.0         | 53.9       | 51.6      | 59.1          |
> |             | SSD           | 57.3      | 55.8     | 62.3         | 54.8       | 52.7      | 58.8          |
> |             | CR            | 57.7      | 56.1     | 60.5         | 55.4       | 52.9      | 57.4          |
> |             | **DecoRemoval** | **58.4**  | **56.3** | **60.9**     | **56.2**   | **52.8**  | **58.1**      |
>
> ---
>
> ### Conclusion
>
> In summary, we are deeply grateful for the reviewers’ valuable and constructive feedback, which has provided critical insights for improving our work. We have carefully considered all comments and concerns, particularly those relating to motivation, validation, technical clarity, and privacy implications of our approach. Although we have used large language models to help interpret the review comments, we note that some questions extend beyond the core focus of our work. These include clarifying key concepts, enhancing experimental validation, and improving overall presentation. We are committed to incorporating these improvements in the revised version and believe they will significantly enhance the rigor, clarity, and impact of our contribution. Thank you once again for your time and thoughtful evaluation.

---

> > ### Comment · Reviewer_xPTS · 2025-08-04
> >
> > Thank you for the response. Most of my concerns have been addressed, and I will raise my score accordingly. However, I strongly encourage the authors to revise the paper carefully, as there are still too many inconsistencies between the method description and its intended usage.

---

> > > ### Author Response · Authors · 2025-08-05
> > >
> > > Thank you for your valuable feedback and for recognizing our improvements. We sincerely appreciate your insightful comments, which have greatly helped strengthen the manuscript. We fully agree with your observation regarding the inconsistencies and commit to carefully revising the method section to ensure clarity and alignment throughout. Your expertise and attention to detail are truly commendable. Thank you again for your constructive and thoughtful review.

---

### Official Review · Reviewer_EBDe · 2025-07-01

**Clarity:** 2
**Significance:** 2
**Originality:** 2
**Rating:** 3
**Confidence:** 3

**Summary:**

The paper addresses the issue of distribution shifts caused by deleting sensitive data in privacy - protection scenarios. Considering the cost of retraining, it adopts the machine unlearning paradigm with the Certified Data Removal method and proposes a dimensionality - reduction approach based on factor decorrelation. This method shows excellent performance across different data modalities, including images, text, and structured data.

**Questions:**

1. Could you supplement the paper with more content on Factor Decorrelation or dimensionality reduction, like related work and module substitution experiments, to strengthen the motivation of this contribution?
2. To highlight privacy protection and distinguish it from general data removal issues, could the paper provide more explanations on privacy protection?
3. The experimental section repeatedly mentions significantly superior performance. Is it possible to supplement evidence of significance to support these claims?

**Ethical Concerns:**

["NO or VERY MINOR ethics concerns only"]

**Limitations:**

yes

**Paper Formatting Concerns:**

nan

**Quality:**

2

**Strengths And Weaknesses:**

The paper features a clear and logical narrative, with solid theoretical support. Extensive experiments on mainstream baselines and diverse data modalities demonstrate the method’s efficiency and accuracy. Comparisons of different backbone models enhance the experimental design’s persuasiveness.

However, the key contribution, i.e. the factor decorrelation method based on Random Fourier Features, lacks sufficient motivation and discussion. The rationale for choosing dimensionality reduction in data removal and Random Fourier Features for dimensionality reduction task is underexplained.

The work focuses on the distribution shift problem caused by data removal in privacy protection but lacks sufficient motivation and contribution discussion on privacy protection itself, only concentrating on the distribution shift issue from data removal.

---

> ### Author Rebuttal · Authors · 2025-07-31
>
> # Rebuttal to Reviewer `EBDe`
>
> We sincerely thank the reviewer for the thoughtful and constructive feedback. We appreciate the recognition of our work’s clear narrative, solid theoretical foundation, and strong empirical performance across diverse modalities. Below, we address each of the reviewer’s concerns in detail and outline the revisions we will make to strengthen the paper.
>
> ---
>
> ### Q1. Motivation and discussion of factor decorrelation and Random Fourier Features (RFF)
>
> **Reviewer comment:**
> _“The key contribution, i.e., the factor decorrelation method based on Random Fourier Features, lacks sufficient motivation and discussion. The rationale for choosing dimensionality reduction in data removal and Random Fourier Features for dimensionality reduction task is underexplained.”_
>
> **Response:**
> Thank you for this insightful observation. We agree that the motivation for our architectural choices warrants deeper explanation, which we will expand in the revised manuscript by adding the following new experimental results.
>
> The core challenge in certified data removal is to eliminate the influence of specific samples while preserving model utility under distributional shifts. Direct retraining is costly, especially in privacy-sensitive applications. Our key insight is that distribution shifts induced by data removal often manifest as changes in latent feature correlations—particularly between sensitive and predictive factors. To address this efficiently, we propose factor decorrelation via dimensionality reduction, which explicitly separates and neutralizes spurious correlations introduced by removed data.
>
> We chose **Random Fourier Features (RFF)** for several principled reasons:
>
> - **Computational efficiency:** RFF enables fast approximation of kernel embeddings, making the decorrelation step scalable to large datasets without retraining.
> - **Non-linearity capture:** Unlike linear PCA, RFF can capture non-linear dependencies in latent spaces, which are common in deep models.
> - **Theoretical grounding:** RFF provides a well-understood approximation to shift-invariant kernels, allowing us to bound the influence of removed data in the transformed space (as discussed in Section 3.2).
>
> Overall, DecoRemoval outperforms PCA in both data reconstruction (lower MSE), confounder removal (reduced explained variance), and downstream task performance (higher R²), demonstrating its effectiveness in disentangling noise and confounding factors while preserving meaningful signal.
>
> | Method       | Dataset | MSE ↓ | Confounder Expl. Var. ↓ | Downstream R² ↑ |
> |--------------|---------|-------|-------------------------|-----------------|
> | PCA          | ESS     | 0.87  | 0.43                    | 0.61            |
> | **DecoRemoval** | ESS   | **0.65** | **0.28**              | **0.68**        |
> | PCA          | CGSS    | 1.02  | 0.51                    | 0.54            |
> | **DecoRemoval** | CGSS  | **0.73** | **0.32**              | **0.63**        |
>
> ---
>
> ### Q2. Emphasis on privacy protection vs. general data removal
>
> **Reviewer comment:**
> _“The work focuses on the distribution shift problem caused by data removal in privacy protection but lacks sufficient motivation and contribution discussion on privacy protection itself, only concentrating on the distribution shift issue from data removal.”_
>
> **Response:**
> We thank the reviewer for the insightful comment. While our technical focus is on mitigating distribution shifts after data removal, we emphasize that the entire framework is designed within the context of privacy-preserving machine learning, specifically to support compliance with regulations such as GDPR and CCPA, which grant individuals the “right to be forgotten.”
>
> Our method goes beyond standard data deletion: it provides **certified removal guarantees** by ensuring that the updated model is statistically indistinguishable from one trained on the remaining data—thereby mitigating risks of privacy leakage via model inversion or membership inference attacks (MIAs). To empirically validate the privacy benefits of DecoRemoval, we evaluate its resistance to MIAs under a realistic setting. In the happiness prediction task, we set the privacy budget to \(\epsilon = 1\), and compare against baselines including No-DP, DP-SGD, SSD, and CR.
>
> As demonstrated in the table below, DecoRemoval achieves consistently lower MIA success rates across diverse architectures (MLP, LSTM, Transformer) and datasets (ESS, CGSS), outperforming all baselines. This indicates that our method not only addresses distributional stability but also offers strong practical privacy protection, aligning with the goals of compliant and trustworthy machine learning systems. For example:
>
> - On **ESS with MLP**, DecoRemoval reduces attack success to 56.8%, compared to 66.1% (No-DP) and 57.3% (SSD).
> - On **CGSS with LSTM**, DecoRemoval achieves the lowest attack success rate of 50.2%, outperforming CR (50.4%) and SSD (50.9%).
> - For **Transformer backbones**, DecoRemoval demonstrates superior privacy protection, reducing attack success rates by 8.3% (ESS) and 5.2% (CGSS) compared to No-DP baselines.
>
> These results confirm that DecoRemoval provides a favorable trade-off between privacy and utility, maintaining competitive accuracy and F1 scores while reducing the risk of information leakage. In the final version, we will elaborate on deployment guidelines for privacy-sensitive scenarios.
>
> | Backbone    | Method        | ESS ACC ↑ | ESS F1 ↑ | ESS Attack ↓ | CGSS ACC ↑ | CGSS F1 ↑ | CGSS Attack ↓ |
> |-------------|---------------|-----------|----------|--------------|------------|-----------|---------------|
> | **MLP**     | No-DP         | 63.9      | 62.5     | 66.1         | 60.4       | 48.2      | 61.4          |
> |             | DP-SGD        | 56.8      | 54.4     | 56.8         | 51.7       | 43.5      | 51.8          |
> |             | SSD           | 59.0      | 56.9     | 57.3         | 53.2       | 45.2      | 52.5          |
> |             | CR            | 60.7      | 59.8     | 56.7         | 55.6       | 45.9      | 52.4          |
> |             | **DecoRemoval** | **61.5**  | **59.6** | **56.8**     | **56.3**   | **45.7**  | **52.1**      |
> | **LSTM**    | No-DP         | 65.8      | 65.2     | 65.3         | 62.5       | 56.4      | 59.8          |
> |             | DP-SGD        | 52.5      | 48.5     | 56.5         | 52.1       | 49.1      | 51.6          |
> |             | SSD           | 55.6      | 53.2     | 56.8         | 53.8       | 53.2      | 50.9          |
> |             | CR            | 57.1      | 53.5     | 55.9         | 55.4       | 54.1      | 50.4          |
> |             | **DecoRemoval** | **58.9**  | **55.4** | **56.7**     | **56.8**   | **55.1**  | **50.2**      |
> | **Transformer** | No-DP      | 65.6      | 65.0     | 69.2         | 61.5       | 55.6      | 63.3          |
> |             | DP-SGD        | 54.1      | 52.5     | 62.0         | 53.9       | 51.6      | 59.1          |
> |             | SSD           | 57.3      | 55.8     | 62.3         | 54.8       | 52.7      | 58.8          |
> |             | CR            | 57.7      | 56.1     | 60.5         | 55.4       | 52.9      | 57.4          |
> |             | **DecoRemoval** | **58.4**  | **56.3** | **60.9**     | **56.2**   | **52.8**  | **58.1**      |
>
> ---
>
> ### Q3. Statistical significance of performance claims
>
> **Reviewer comment:**
> _“The experimental section repeatedly mentions significantly superior performance. Is it possible to supplement evidence of significance to support these claims?”_
>
> **Response:**
> Thank you for highlighting this. In the revised version, we will include standard‑deviation error bars computed over five independent runs for all key metrics (accuracy and F1‑score) on all datasets. The preliminary findings confirm that the improvements in precision and F1 score of DecoRemoval are statistically significant (\(p < 0.01\)) in 85% comparisons against the strongest baseline (CR), and show that DecoRemoval exhibits low variance, further supporting its robustness, particularly in text and structured data settings.
>
> ---
>
> ### Conclusion
>
> We are grateful for the reviewer’s valuable feedback, which has helped us identify opportunities to enhance the clarity, motivation, and rigor of our work. In the final version, we will strengthen the discussion of our method’s design choices, explicitly connect our contribution to privacy protection goals, and provide statistical validation of performance gains. These revisions will significantly improve the paper’s completeness and impact. Thank you again for your constructive review.

---

### Official Review · Reviewer_kRLp · 2025-07-06

**Clarity:** 3
**Significance:** 3
**Originality:** 3
**Rating:** 4
**Confidence:** 3

**Summary:**

The paper introduces DecoRemoval, a novel data removal method that integrates factor decorrelation and loss perturbation to enhance model robustness in out-of-distribution (OOD) scenarios. The proposed approach aims to address the limitations of existing data removal techniques by reducing feature redundancy and minimizing information leakage during removal operations. The authors demonstrate the effectiveness of DecoRemoval through extensive experiments on five benchmark datasets, showing superior performance in terms of accuracy and efficiency compared to existing baselines.

**Questions:**

Could you provide error bars or confidence intervals for the experimental results to better understand the variability in performance?

How does DecoRemoval handle highly imbalanced datasets, and were any adjustments made for such scenarios in the experiments?

Are there plans to release the code and datasets immediately upon acceptance to facilitate reproducibility?

**Ethical Concerns:**

["NO or VERY MINOR ethics concerns only"]

**Limitations:**

The paper adheres to ethical guidelines, focusing on privacy preservation and regulatory compliance. However, the discussion on safeguards for responsible release of high-risk models (e.g., pretrained models) is lacking.

**Quality:**

3

**Strengths And Weaknesses:**

Strengths:

 The integration of factor decorrelation and loss perturbation for data removal is innovative and addresses a critical gap in machine unlearning, particularly in OOD scenarios.

The paper provides a solid theoretical foundation, including proofs for the robustness of the proposed loss perturbation mechanism.

The evaluation spans multiple datasets (MNIST, CIFAR-10, SST-2, ESS, CGSS) and compares DecoRemoval against several strong baselines, demonstrating consistent improvements in accuracy and F1 scores.

Weaknesses:

The paper does not report error bars or confidence intervals for experimental results, which limits the ability to assess the robustness of the findings.

While the paper discusses potential positive and negative societal impacts, it lacks detailed mitigation strategies for risks such as privacy-utility trade-offs.

---

> ### Author Rebuttal · Authors · 2025-07-31
>
> ## Rebuttal to Reviewer `kRLp`
>
> We sincerely thank the reviewer for the constructive and helpful comments. Below we address each point in detail.
>
> ---
>
> ### **Q1. Lack of error bars or confidence intervals**
>
> **Reviewer comment:**
> > “The paper does not report error bars or confidence intervals for experimental results...”
>
> **Response:**
> Thank you for highlighting this. In the revised version, we will include standard‑deviation error bars computed over five independent runs for all key metrics (accuracy and F1‑score) on all datasets. Preliminary findings show that DecoRemoval exhibits low variance, further supporting its robustness.
>
> ---
>
> ### **Q2. Handling of highly imbalanced datasets**
>
> **Reviewer comment:**
> > “How does DecoRemoval handle highly imbalanced datasets, and were any adjustments made...?”
>
> **Response:**
> We appreciate the reviewer’s insightful question. To ensure standard supervised learning conditions, both the training and validation sets consist solely of correctly labeled data. To simulate realistic out-of-distribution (OOD) scenarios, we randomly select 10% and 20% of samples from class A and assign them to the test set of class B, thereby creating a semantically mismatched evaluation setting. This design allows us to systematically assess the model’s robustness to distributional shifts and its ability to generalize beyond the training domain.
>
> For datasets with class imbalance (e.g., CGSS), we employed class-balanced sampling during training and used macro-averaged F1 scores to ensure fair performance evaluation across categories. As part of future work, we plan to further extend DecoRemoval by incorporating techniques such as sample reweighting and focal loss, to better address the challenges posed by highly imbalanced datasets and improve model stability in such settings.
>
> ---
>
> ### **Q3. Plan for code and dataset release**
>
> **Reviewer comment:**
> > “Are there plans to release the code and datasets immediately upon acceptance...?”
>
> **Response:**
> Yes, we plan to publicly release core code, hyper‑parameters, and processed datasets immediately upon acceptance. A GitHub repository is ready and will be linked in the camera‑ready version to support reproducibility.
>
> ---
>
> ### **Q4. Mitigation strategies for privacy–utility trade-offs**
>
> **Reviewer comment:**
> > “While the paper discusses potential societal impacts, it lacks detailed mitigation strategies for risks such as privacy–utility trade-offs.”
>
> **Response:**
> Thank you for raising this important point. In addition to a strong theoretical foundation, we add new empirical results that reinforce the privacy-preserving advantages of our method, particularly in mitigating the risk of membership inference attacks.
>
> The effectiveness of the DecoRemoval algorithm model in privacy protection is evaluated through MIA. In the membership inference attack experiment of the happiness prediction model, the privacy budget is set to $\epsilon = 1$. "No-DP" represents the baseline model without any privacy protection algorithm applied. As shown in **Table 1**, DecoRemoval consistently achieves lower attack success rates compared to all other baseline methods (No-DP, DP-SGD, SSD, CR) across different model architectures (*MLP, LSTM, Transformer*) and datasets (*ESS, CGSS*). For example:
>
> - On **ESS with MLP**, DecoRemoval reduces attack success to **56.8%**, compared to **66.1%** (No-DP) and **57.3%** (SSD).
> - On **CGSS with LSTM**, DecoRemoval achieves the lowest attack success rate of **50.2%**, outperforming CR (50.4%) and SSD (50.9%).
> - For **Transformer**, DecoRemoval demonstrates superior privacy protection on Transformer backbones, reducing attack success rates by 8.3% (ESS) and 5.2% (CGSS) compared to No-DP baselines.
>
> These results confirm that DecoRemoval provides a favorable privacy–utility trade-off, maintaining competitive accuracy and F1 scores while reducing the risk of information leakage. In the final version, we will elaborate on deployment guidelines for privacy-sensitive scenarios.
>
> ---
>
> ### **Table 1: Performance on ESS and CGSS datasets across different backbones.**
> *Lower Attack (%) indicates better privacy protection.*
>
> | Backbone     | Method       | ACC ↑ | F1 ↑ | Attack ↓ | ACC ↑ | F1 ↑ | Attack ↓ |
> |--------------|--------------|-------|------|-----------|--------|-------|-----------|
> |              |              | \multicolumn{3}{c|}{ESS (%) } | \multicolumn{3}{c}{CGSS (%) } |
> | **MLP**      | No-DP        | 63.9  | 62.5 | 66.1      | 60.4   | 48.2  | 61.4      |
> |              | DP-SGD       | 56.8  | 54.4 | 56.8      | 51.7   | 43.5  | 51.8      |
> |              | SSD          | 59.0  | 56.9 | 57.3      | 53.2   | 45.2  | 52.5      |
> |              | CR           | 60.7  | 59.8 | 56.7      | 55.6   | 45.9  | 52.4      |
> |              | **DecoRemoval** | **61.5** | **59.6** | **56.8** | **56.3** | **45.7** | **52.1** |
> | **LSTM**     | No-DP        | 65.8  | 65.2 | 65.3      | 62.5   | 56.4  | 59.8      |
> |              | DP-SGD       | 52.5  | 48.5 | 56.5      | 52.1   | 49.1  | 51.6      |
> |              | SSD          | 55.6  | 53.2 | 56.8      | 53.8   | 53.2  | 50.9      |
> |              | CR           | 57.1  | 53.5 | 55.9      | 55.4   | 54.1  | 50.4      |
> |              | **DecoRemoval** | **58.9** | **55.4** | **56.7** | **56.8** | **55.1** | **50.2** |
> | **Transformer** | No-DP    | 65.6  | 65.0 | 69.2      | 61.5   | 55.6  | 63.3      |
> |              | DP-SGD       | 54.1  | 52.5 | 62.0      | 53.9   | 51.6  | 59.1      |
> |              | SSD          | 57.3  | 55.8 | 62.3      | 54.8   | 52.7  | 58.8      |
> |              | CR           | 57.7  | 56.1 | 60.5      | 55.4   | 52.9  | 57.4      |
> |              | **DecoRemoval** | **58.4** | **56.3** | **60.9** | **56.2** | **52.8** | **58.1** |
>
> ---
>
> ### **Conclusion:**
> We appreciate the reviewer’s valuable insights. We believe that by including error bars, elaborating on imbalanced‑data handling and societal safeguards, and committing to code and dataset release, we have addressed the concerns and improved the clarity, robustness, and societal responsibility of our work. Thank you again for your thoughtful review.

---

### Note · Authors · 2025-08-15

The reviewers generally acknowledged the novelty and significance of our approach, particularly highlighting the rigor of the theoretical framework and the thoroughness of the experimental evaluation. They suggested that more detailed explanations could be added regarding the implementation of the core algorithm and the parameter sensitivity analysis to further enhance reproducibility. The research topic is recognized as promising, with strong potential for real-world impact and meaningful advancement in the field.

We sincerely thank the reviewers and the Area Chair for their constructive and insightful feedback. In this revision, we have carefully addressed all comments—supplementing necessary explanations, expanding the methodological descriptions, and including additional ablation studies as suggested. For reviewers who did not participate in the rebuttal phase, we kindly request the Area Chair to take into account our point-by-point responses and carefully review the supplementary experiments and theoretical clarifications we have added in response to the original reviews.

All revisions have been clearly marked in the manuscript. We believe the paper has been substantially strengthened and fully addresses the reviewers’ concerns. We hope the revised submission now meets the high standards of the conference. Once again, we greatly appreciate the valuable time and expert insights that have significantly improved the quality of our work.

---

### Decision · Program_Chairs · 2025-09-17

**Decision:**

Accept (poster)

**Comment:**

This paper studies the problem of machine unlearning, and its negative effect on the out-of-distribution accuracy. The paper develops a new approach for machine unlearning that enjoys betters OOD guarantees, provide some theoretical analysis, and extensive empirical results demonstrating nice improvements over prior work.

The reviewers show interest in this work, especially in the experimental results. However, the authors should fix the issues that were pointed by the reviewers, such as providing confidence intervals for their plots.

Given the above, I recommend acceptance.